# A Fractional-Order Multi-Rate Repetitive Controller for Single-Phase Grid-Connected Inverters

**Qiangsong Zhao** [1], **Kaiyue Liu** [1,*] **and Hengyi Li** [2]

1    School of Electronic and Information, Zhongyuan University of Technology, Zhengzhou 451191, China
2    Graduate School of Science and Engineering, Ritsumeikan University, Kusatsu 525-0058, Japan
*    Correspondence: liukaiyue@zut.edu.cn

**Abstract:** The multi-rate repetitive controller (MRC) can achieve zero steady-state error in tracking the reference current signal of grid-connected inverters, save the settling time effectively, and improve the running speed. However, when the grid frequency fluctuates, the harmonic suppression performance of MRC will degrade. Aiming at the problem of harmonic suppression performance degradation, a fractional-order MRC (FOMRC) based on the farrow structure fractional delay (FD) filter is proposed. Firstly, the equivalent digital model of MRC is established, and a Farrow structure fractional delay (FD) filter based on Taylor series expansion is selected as the internal model filter of MRC. The stability analysis and harmonic suppression characteristics of the FOMRC are analyzed. Then, the parameter design of FOMRC applied to an LCL single-phase grid-connected inverter control system is given. Finally, the simulation results show that the proposed method has better transient and steady-state performance than the CRC when the grid frequency fluctuates.

**Keywords:** repetitive controller; multi-rate; fractional delay

## 1. Introduction

With the proposal of a carbon peak and carbon neutral target in our country, the step of clean and low-carbon transition of the power system will be further accelerated. According to various policy orientations and the future direction of clean energy development, distributed energy will have a gradual growth and may become a major energy source in the future. Distributed energy is a kind of clean energy with strong adaptability to the environment. However, many non-linear power electronic switching devices are used in distributed generation, which will inject a large amount of harmonic current into the grid system and reduce the power quality [1]. The grid-connected inverter is the interface unit of power transmission from distributed generation system to the power grid, and its performance directly determines the quality of grid-connected current. According to the IEEE standard, the total harmonic distortion (THD) of grid-connected current must be less than 5%.

In order to reduce the harmonic content of the grid current of the inverter, L-type filter, and LCL-type filter are usually used to suppress the high-frequency harmonic at the switching frequency of the grid current. Under the same harmonic suppression requirements, the LCL filter is widely used because of its smaller actual volume and lower cost, which can better suppress grid current harmonics [2,3]. For the suppression of low-frequency harmonics in grid current, many controllers have been proposed. Among them, the commonly used include proportional integral (PI) controller [4], proportional resonant (PR) controller [5], proportional multi-resonant (PMR) controller, repetitive controller (RC), etc.

The repetitive controller (RC) [6–9] is widely used because it can realize the static error-free tracking and disturbance rejection of fundamental and harmonics of sinusoidal signals. However, when the conventional repetitive controller (CRC) is implemented digitally, the sampling rate of the repetitive controller and the switching frequency of

the inverter power device are often 10 kHz or even higher, which makes the memory consumption of the controller too large and the calculation burden too heavy. For this reason, multiple rate repetitive controller (MRC) [10,11] has been proposed, in which the repetitive controller is set in a low sampling frequency environment, and the feedback part of the grid-connected current still adopts a high sampling rate. Compared with CRC, the memory consumption and calculation times of the MRC controller in each sampling period are greatly reduced [12]. Previous work has shown that MRC can maintain the same convergence rate and total harmonic distortion (THD) of grid-connected current as CRC, and CRC is a special case when MRC sampling factor is 1.

However, in the practical application of inverters, there is usually a small range fluctuation of the power grid frequency, which makes the internal mode of the RC periodic signal not match with the periodic signal to be tracked or eliminated, resulting in the attenuation of the resonant frequency gain [13,14]. The internal mode delay number $N = f_s/f_g$, that is, the ratio of the sampling frequency to the fundamental frequency. When grid frequency fluctuates, $N$ and the reduced-sampled $N$ may be a fraction, and fractional delay can not be achieved directly in digital systems, it will reduce the repetitive control gain and the harmonic suppression ability of the system. The variable sampling frequency method [15] can ensure that $N$ is an integer to achieve appropriate harmonic suppression. However, the variable sampling rate means that the change of system dynamics, especially the change of the controlled object model, increases the difficulty of system stability analysis. The finite impulse response (FIR) filter based on the Lagrange interpolation method is widely used as the internal model filter of various repetitive controllers (odd RC [16], dual-mode RC [17], selective harmonic RC [18]) because of its simple structure and linear phase in the whole frequency band. The fractional delay link (FD) is approximated to obtain any numerical delay $N$, which improves the anti-frequency fluctuation ability of repetitive control. However, the FIR filter based on Lagrange interpolation needs to recalculate the coefficients of $M + 1$ ($M$ is the order of the FIR filter) sub-filters when the coefficients are updated, which greatly increases the amount of calculation and is slightly contrary to the original intention of using MRC in this paper.

Therefore, this paper proposes a fractional-order multi-rate repetitive controller (FOMRC) composed of a Farrow structure FD filter based on Taylor series expansion. FOMRC consists of an MRC with a low sampling rate and a relatively simple Farrow structure FD filter. Each sub-filter of the FD filter with a Farrow structure is designed offline in advance. For the new FD, only the fractional value of the fractional $N$ is adjusted, which greatly reduces the computational burden of the controller. At present, the FOMRC remains to be further studied. This composite controller can not only reduce the computational burden of the digital system and retain the harmonic suppression ability of the CRC but can also accurately approximate the FD generated when the fundamental frequency of the power grid fluctuates. Improve the steady-state accuracy of multi-rate repetitive control.

## 2. Fractional Order Multi-Rate Repetitive Controller

### 2.1. MRC

Figure 1 shows the block diagram of a conventional plug-in MRC system. In the figure, the sampling period of the grid current feedback system is $T_s$, $E(z)$ is the error between the input signal $R(z)$ and the output signal $Y(z)$. In order to prevent the distortion of $E(z)$ signal outside the Nyquist frequency when entering the repetitive controller with a low sampling rate, the anti-spectral aliasing filter $F_1(z)$ is often used to intercept the signal in the $|\omega| > \pi/T_m$ part, and the signal $E(z_m)$ is obtained by downsampling as the input of RC. The output $U_r(z_m)$ of RC is up-sampled by zero-order holder (ZOH) to obtain $U_r(z)$, and then an anti-imaging filter $F_2(z)$ with cutoff frequency less than $\pi/T_m$ is connected to prevent distortion. $G_{pi}(z)$ is a proportional integral controller for stabilizing the closed-loop control system. $P(z)$ is the controlled object, and $D(z)$ is the disturbance source.

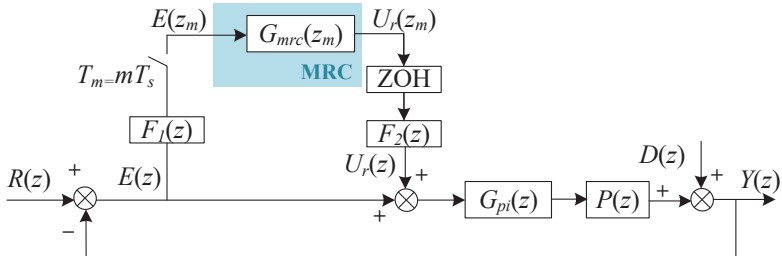

**Figure 1.** Plug-in MRC system.

The sampling period of RC is $T_m$, which is $m$ times of $T_s$, and $m$ is the sampling factor. The relationship between the two sampling rates is

$$T_m = mT_s; \quad z = e^{sT_s}; \quad z_m = e^{sT_m} = z^m \tag{1}$$

In order to simplify the analysis, the multi-rate system is transformed into an equivalent single-rate system with a low sampling rate, and the structure is shown in Figure 2. $G_{pi}(z_m)$ and $P(z_m)$ are the equivalent transfer functions of PI controller and controlled object at low rates, respectively; $k_r$ is the internal mode gain; $Q(z_m)$ is used to improve the robustness of RC; $N_m = T_g/T_m$, $T_g$ is the fundamental frequency period of the reference signal and disturbance signal. $S(z_m)$ is a low-pass filter used to accelerate the high-frequency amplitude attenuation of the controlled object; $z_m^k$ is the phase-lead compensation, which is used to compensate the phase lag of the control system. The influence of $F_1(z)$, downsampling link, zero order holder and $F_2(z)$ on the system can also be compensated by the $z_m^k$.

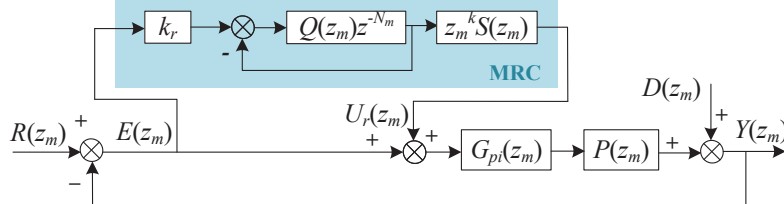

**Figure 2.** Equivalent single-rate control system.

The transfer function of MRC is

$$G_{mrc} = k_r \frac{z_m^{-N_m} Q(z_m)}{1 - z_m^{-N_m} Q(z_m)} S(z_m) z_m^k \tag{2}$$

*2.2. Farrow Structure FD Filter Design Based on Taylor Series*

When $N_m$ is a fraction, $N_m$ can be divided into integer $D$ and fractional $d$, $z_m^{-N_m} = z_m^{-D} z_m^{-d}$. The FD transfer function $z_m^{-d}$ can be expanded to a polynomial about $d$ by the Taylor series [19]:

$$a_n(d) = z_m^{-d} = e^{-j\omega md} = \sum_{k=0}^{M} \frac{(-j\omega m)^k}{k!} (d)^k \tag{3}$$

where, $0 \leq d < 1$.

The FD filter can be expressed as

$$G_d(z_m) = \sum_{n=0}^{S} a_n(d) z_m^{-n} \tag{4}$$

where $S$ is the order of the FD filter and $a_n(d)$ is the polynomial of $d$.

Substituting (3) into (4), the Farrow [20] FD filter $G_d(z_m)$ can be obtained

$$G_d(z_m) = \sum_{k=0}^{M} \sum_{n=0}^{S} a_{nk} z_m^{-n} d^k = \sum_{k=0}^{M} L_k(z_m) d^k \tag{5}$$

$L_k(z_m)$ ($k = 0,1, \dots , M$) is the $k$ subfilter in $G_d(z_m)$.

Usually choosing $S = M$, subfilter calculation formula based on Lagrange interpolation method is as follows [21,22]:

$$U = \begin{bmatrix} 0^0 & 0^1 & 0^2 & \cdots & 0^S \\ 1^0 & 1^1 & 1^2 & \cdots & 1^S \\ 2^0 & 2^1 & 2^2 & \cdots & 2^S \\ \vdots & \vdots & \vdots & \vdots & \vdots \\ M^0 & M^1 & M^2 & \cdots & M^S \end{bmatrix} \tag{6}$$

$$z_{sub} = \begin{bmatrix} 1 & z_m^{-1} & z_m^{-2} & \cdots & z_m^{-M} \end{bmatrix}^T \tag{7}$$

$$f_{sub} = U^{-1} z_{sub} = \begin{bmatrix} L_0(z_m) & L_1(z_m) & L_2(z_m) & \cdots & L_M(z_m) \end{bmatrix}^T \tag{8}$$

In the formula, $f_{sub}$ is a subfilter matrix, $z_{sub}$ is a delay operator matrix, and $U$ is a Vandermon matrix. For different fractional delay $z_m^{-d}$, the amplitude response of the first and second-order FD filters based on the Lagrange interpolation method is shown in Figure 3. It can be seen that the bandwidth of the first-order FD filter is 50% of the Nyquist frequency, and the bandwidth of the second-order FD filter is 63.5% of the Nyquist frequency. It shows that the FD filter based on the Taylor series expansion Farrow structure can effectively approximate the fractional delay $z_m^{-d}$. Generally, the first-order FD is sufficient to provide sufficient bandwidth to compensate for harmonic distortion, but higher-order FD can further reduce the error. In this paper, the order $M = 2$.

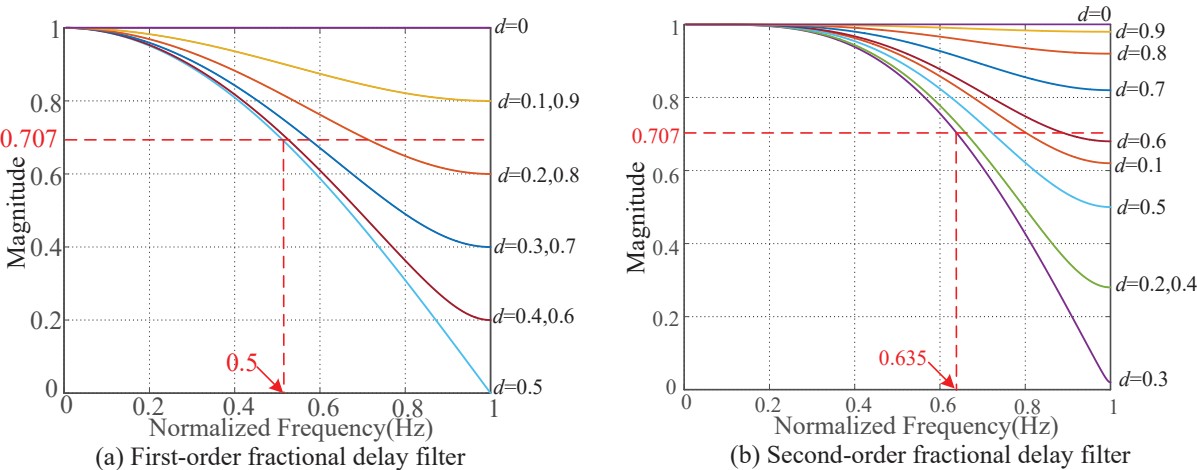

**Figure 3.** Amplitude response of first-order and second-order FD filters based on Taylor series expansion.

Thus, the transfer function of FOMRC consisting of the Farrow structured FD filter based on the Taylor series is

$$G_{fomrc} = k_r \frac{z_m^{-D} G_d(z_m) Q(z_m)}{1 - z_m^{-D} G_d(z_m) Q(z_m)} S(z_m) z_m^k \tag{9}$$

### 2.3. Stability Analysis of FOMRC

The steady-state error of the FOMRC system is obtained from Figure 2.

$$E(z_{\mathrm{m}}) = \frac{\left(i_{ref}(z_m) - u_g(z_m)\right)}{1 + (G_{rc}(z_m) + 1)G_{op}(z_m)}$$
$$= \frac{\left(i_{ref}(z_m) - u_g(z_m)\right)\left(1 + G_{pi}(z_m)P(z_m)\right)^{-1}\left(1 - z_m^{-D}G_d(z_m)Q(z_m)\right)}{\left(1 - z_m^{-D}G_d(z_m)Q(z_m)\left(1 - k_r z_m^k S(z)G_{cl}(z_m)\right)\right)} \quad (10)$$

where, $G_{cl}(z_m) = G_{pi}(z_m)P(z_m)/(1 + G_{pi}(z_m)P(z_m))$, is the closed-loop transfer function of the system before RC is inserted. The FOMRC system is stable when the following two conditions are satisfied

① The roots of the $1 + G_{\mathrm{pi}}(z_m)P(z_m) = 0$ are in the unit circle.;

② $\left|z_m^{-D}G_d(z_m)Q(z_m)\left(1 - k_r z_m^k S(z_m)G_{cl}(z_m)\right)\right| < 1, \forall z = e^{j\omega T_m}, 0 < \omega < \pi/T_m$.

It is known that stability condition 1 has nothing to do with FD. For condition 2,

$$\left|Q(z_m)\left(1 - k_r z_m^k S(z_m)G_{cl}(z_m)\right)\right| < |z_m^{-D}G_d(z_m)|^{-1}, \forall z = e^{j\omega T_m}, 0 < \omega < \pi/T_m \quad (11)$$

Within the bandwidth of the FD filter, $\left|z_m^{-D}G_d(z_m)\right|^{-1} \to 1$, the stability of the system is independent of the FD filter.

### 2.4. Analysis of Harmonic Suppression Characteristics of FOMRC

When the sampling frequency is fixed and the grid frequency changes, $N_m$ may be a fraction. Taking the sampling factor $m = 2$ as an example, when the fundamental frequency $f_g$ changes from 50 Hz to 49.6 Hz and 50.4 Hz, the internal mode frequency characteristics of MRC ($N_m = 100$) and FOMRC ($N_m = 100.8, 99.2$) at the 8th harmonic frequency are shown in Figure 4. The open-loop gain of MRC decreased from 36.07 dB to 13.85 dB when $f_g$ changed from 50 Hz to 49.6 Hz, while FOMRC still provided higher open-loop gain, which was not affected by frequency fluctuation and had better harmonic suppression effect.

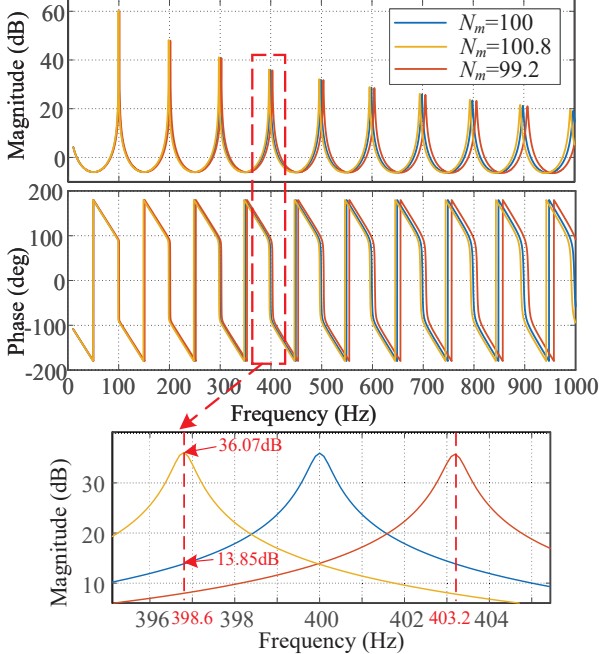

**Figure 4.** The frequency characteristics of MRC ($N_m = 100$) and FOMRC ($N_m = 100.8, 99.2$).

### 3. Modeling and Parameter Design of LCL Single-Phase Grid-Connected Inverter

As shown in Figure 5, the LCL single-phase grid-connected inverter is used as the control object for parameter design. $E_d$ is dc bus voltage, $u_i$ is inverter output voltage. $u_g$ is the grid voltage, $i_g$ is the grid current, and $i_{ref}$ is the reference current. $L_1$ and $L_2$ are filter inductance, $C$ is the filter capacitor, and $R$ is the series resistance of the capacitor to suppress the resonance peak generated by LCL filter.

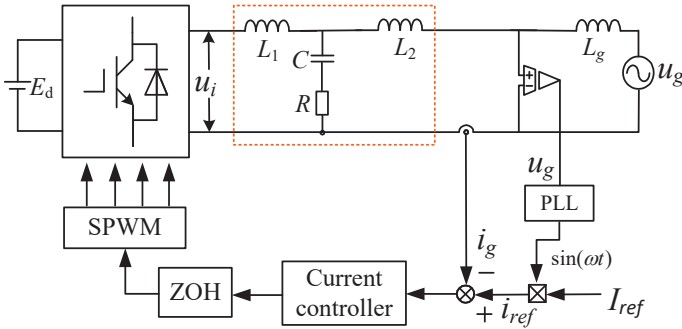

**Figure 5.** LCL-type single-phase grid-connected inverter model.

Regardless of the equivalent resistance of the inductor, the transfer function from the input voltage $u_i$ to the grid current $i_g$ is

$$P(s) = \frac{CRs + 1}{CL_1L_2s^3 + C(L_1 + L_2)Rs^2 + (L_1 + L_2)s} \tag{12}$$

Table 1 shows the parameters of an LCL-type single-phase grid-connected inverter. With parameters, (12) can be obtained by discretization

$$P(z) = \frac{0.006135z^2 + 0.004307z - 0.002401}{z^3 - 2.005z^2 + 1.493z - 0.4879} \tag{13}$$

**Table 1.** Parameters of the LCL-type single-phase grid-connected inverter.

| Parameters | Value |
|---|---|
| Inverter side inductance:$L_1$ | 3.8 mH |
| Grid side inductance:$L_2$ | 2.2 mH |
| Filter capacitance:$C$ | 10 uF |
| Passive damping resistor:$R$ | 10 Ω |
| Dc bus voltage:$E_d$ | 380 V |
| Grid frequency:$f_g$ | 50 Hz |
| Sampling frequency:$f_s$ | 10 kHz |
| Switching frequency:$f_{sw}$ | 10 kHz |
| Switch dead time: | 3 us |
| Sampling ratio: | 2 |

Considering that the sampling factor $m$ is too large, the sampling frequency of the repetitive controller will be too low, which will result in insufficient or over-compensation of phase, and affect the system stability and harmonic suppression accuracy. Therefore, the sampling factor $m$ is set to 2. Select filter $F_1(z) = F_2(z) = 0.15z^{-1} + 0.7 + 0.15z$, the cutoff frequency is $2\pi^*2460 < 2\pi^*2500 = \pi/T_m$.

According to [12], the parameters of the PI controller are designed and calculated. $k_p = 10, k_i = 1300$. $G_{cl}(z_m)$ has a amplitude margin of 9.95 dB and a phase margin of $154°.Q(z_m)$ usually selects the low-pass filter with a gain of less than 1 or a constant of less than 1 to improve the stability of RC. In this paper, $Q(z_m) = 0.25z_m + 0.5 + 0.25z_m^{-1}$. In order to make $G_{cl}(z_m)$ have faster amplitude attenuation at high frequencies, select

the fourth-order Butterworth low-pass filter $S(z_m)$ with a cut-off frequency of 1kHz. The expression of $S(z_m)$ is as follows

$$S(z_m) = \frac{0.04658z_m^4 + 0.1863z_m^3 + 0.2795z_m^2 + 0.1863z_m + 0.04658}{z_m^4 - 0.7812z_m^3 + 0.68z_m^2 - 0.1827z_m + 0.03012} \tag{14}$$

In order to determine the phase lead order $k$, the Bode diagram of $z_m^k S(z_m)G_{cl}(z_m)$ at $3, 4, 5, 6$ is given in Figure 6. It can be seen that when $k = 4$, the phase of $S(z_m)G_{cl}(z_m)$ is close to $0°$ within 1 kHz, and the compensation effect is the best.

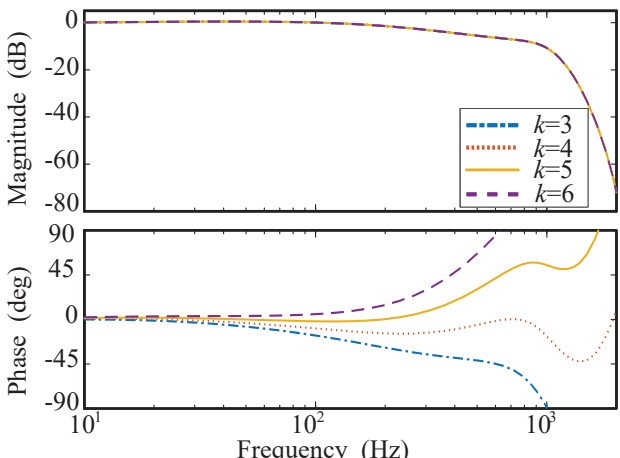

**Figure 6.** The amplitude-frequency and phase-frequency characteristics of $S(z_m)G_{cl}(z_m)$ when $k$ takes different values.

Finally, select $k_r$ according to condition 2. According to (11)

$$\left| Q(z_m)\left(1 - k_r z_m^k S(z_m)G_{cl}(z_m)\right) \right| < 1, \forall z = e^{j\omega T_m}, 0 < \omega < \pi/T_m \tag{15}$$

Define $H(z_m) = Q(z_m)\left[1 - k_r z_m^k S(z_m)G_{cl}(z_m)\right], \quad z_m = e^{j\omega T_m}$. Then when the locus of $H(e^{j\omega T_m})$ is in the unit circle, the system is stable. The internal model gain $k_r$ directly determines the error attenuation speed of the repetitive control system, and also affects the Nyquist curve of $H(e^{j\omega T_m})$. It can be seen from Figure 7 that when $k_r \leq 1.5$, the trajectory of $H(e^{j\omega T_m})$ is always in the unit circle, and there is still a certain distance between the trajectory and the boundary of unit circle, indicating that the system has sufficient stability margin. The direction of the arrow represents the trajectory direction of the amplitude curve of $H(e^{j\omega T_m})$ from 0 Hz to Nyquist frequency. When $k_r$ is 0.5, 1, 1.5, and 2, respectively, the corresponding points of the amplitude of $H(e^{j\omega T_m})$ at 150 Hz and 1000 Hz are shown in the figure. It can be seen that when $k_r = 1$, the low frequency region of $H(e^{j\omega T_m})$ is closer to the center of the unit circle, and the controller has better low-frequency signal tracking ability in the low frequency region.

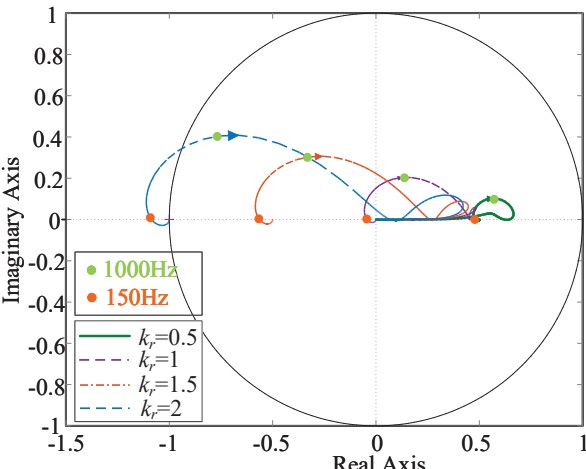

**Figure 7.** The trajectory of $H(e^{j\omega T_m})$ when $k_r$ takes different values.

## 4. Simulation Analysis

In order to verify the performance of the FOMRC system, the LCL-type single-phase grid-connected inverter model is built in MATLAB / Simulink simulation environment, and the steady-state and dynamic performance are compared with CRC. The reference current is 10 A. The CRC system, that is, sampling factor $m = 1$, sampling frequency $f_s = 10$ kHz, delay beat $N = f_s/f_g$, phase lead order $k = 8$, other parameters remain unchanged.

### 4.1. Steady-State Performance

As can be seen from Figure 8, when $f_g = 50$ Hz, $N = T_g/T_s = 200$, $N_m = T_g/T_m = 100$, the output voltage and current waveforms of CRC and FOMRC are almost sinusoidal, the THD of CRC and FOMRC are 0.97% and 1.33%, respectively, both of them can effectively suppress the harmonic in the power grid.

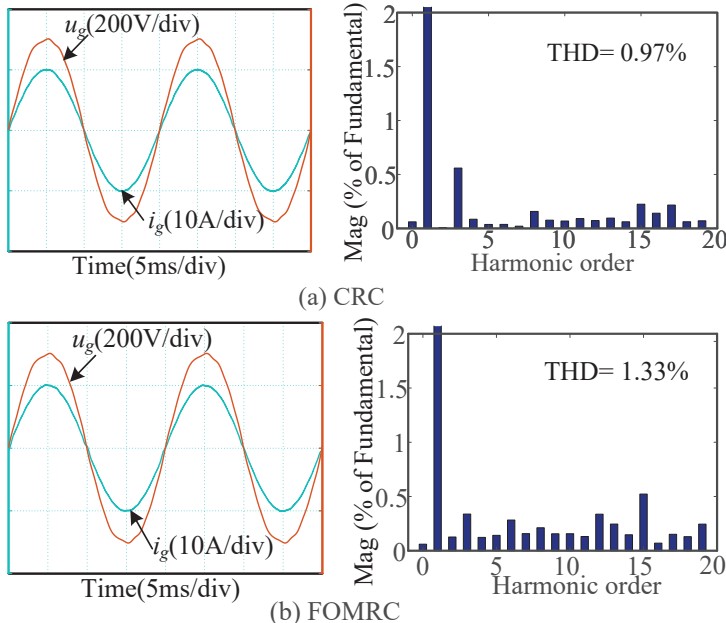

**Figure 8.** Voltage and current waveforms and output current spectrum of the two control systems with integer delay ($f_g = 50$ Hz).

Figures 9 and 10 show that when $f_g = 49.6$ Hz, $N = T_g/T_s = 201.6$, $N_m = T_g/T_m = 100.8$; when $f_g = 50.4$ Hz, $N = T_g/T_s = 198.4$, $N_m = T_g/T_m = 99.2$. The voltage and

current waveform and output current spectrum of CRC and FOMRC control systems under fractional delay are obtained. When $f_g = 49.6$ Hz, the THD of CRC and FOMRC are 2.71% and 1.26%, respectively. When $f_g = 50.4$ Hz, the THD of CRC and FOMRC are 3.33% and 1.19%, respectively. It can be seen that when the fractional delay occurs, the output current waveform of CRC is seriously distorted and the harmonic content is high. The output current waveform of FOMRC is still a smooth sine curve, and the harmonic suppression ability is significantly improved compared with CRC.

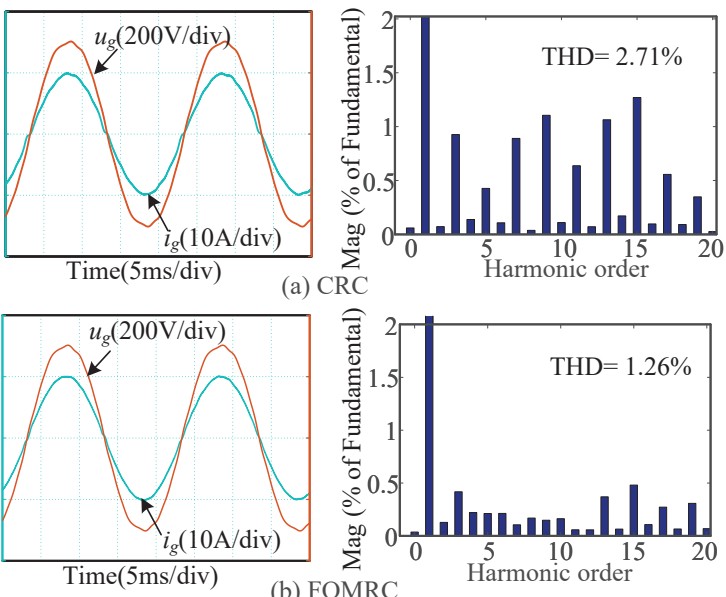

**Figure 9.** Voltage and current waveforms and current spectrum analysis of the two control systems at the fractional delay ($f_g = 49.6$ Hz).

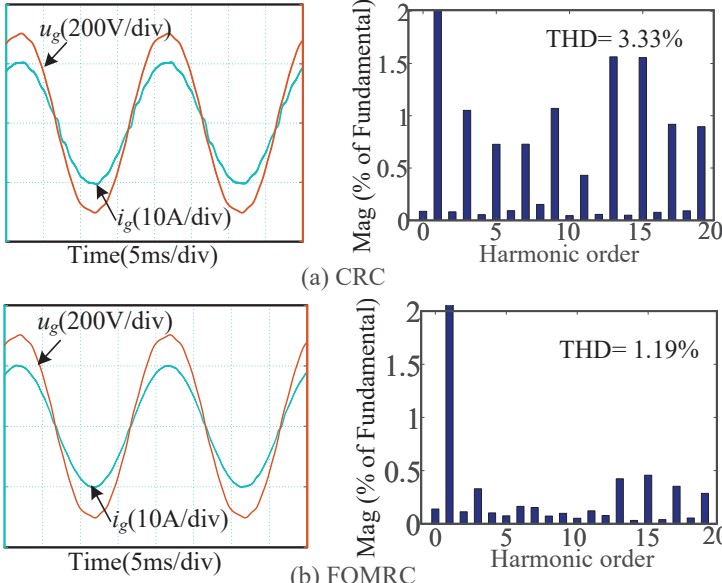

**Figure 10.** Voltage and current waveforms and current spectrum analysis of the two control systems at the fractional delay ($f_g = 50.4$ Hz).

Currently, a number of scholars have used a third-order FIR filter based on Lagrange interpolation to approximate the fractional delay [16–18] and achieved good harmonic suppression results. To verify that the FOMRC proposed in this paper has similar performance, Figure 11 gives a comparison plot of the THD results of CRC, MRC, and MRC using FIR to approximate the fractional delay, and the FOMRC proposed in this paper when the grid

frequency fluctuates in the range of $50 \pm 0.4$ Hz. It can be seen that the THD of MRC is slightly reduced compared with CRC due to the halving of the sampling rate when the grid frequency is 50 Hz. The larger the grid frequency offset, the more serious the degradation of the harmonic rejection ability of MRC. The proposed method can obtain smaller THD values, better frequency adaptivity, and the same anti-frequency fluctuation capability as the method using FIR filter with approximate fractional delay.

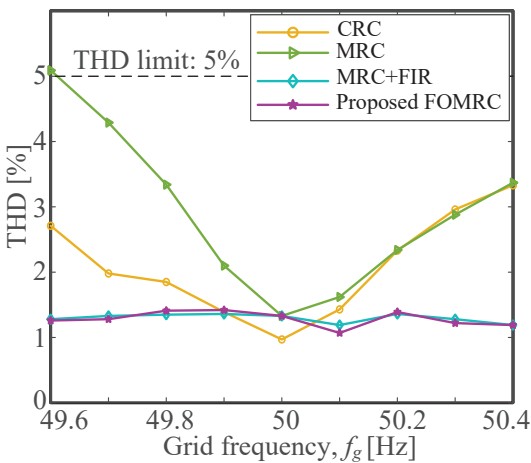

**Figure 11.** Comparison of THD results for the above control systems.

### 4.2. Dynamic Performance

Figures 12 and 13 show the dynamic performance of CRC and FOMRC control systems with reference current amplitude reduced from 10 A to 6 A at $t$ = 0.25 s. When the fundamental frequency is 49.6 Hz, the current error of the CRC system is ±1 A, and that of the FOMRC system is ±0.3 A. When the fundamental frequency is 50.4 Hz, the current error of the CRC system is ±1.1 A, and that of the FOMRC system is ±0.3 A. The errors of the two control systems can converge again in two fundamental periods when the current changes abruptly at different frequencies. It can be seen that FOMRC and CRC have similar error convergence speeds when the power grid frequency fluctuates, but can effectively reduce the error.

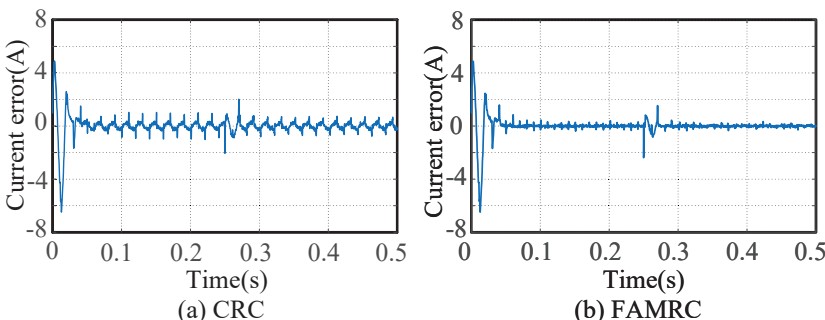

**Figure 12.** Current error of two control systems with grid frequency fluctuation ($f_g$ = 49.6 Hz).

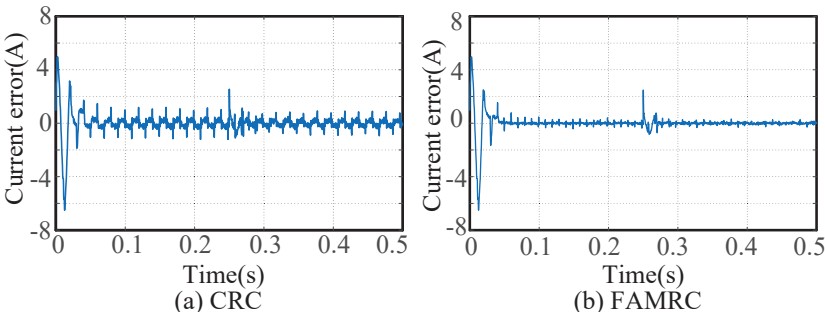

**Figure 13.** Current error of two control systems with grid frequency fluctuation ($f_g$ = 50.4 Hz).

## 5. Conclusions

When the grid frequency fluctuates, the performance of the grid-connected inverter based on MRC will decrease. Therefore, this paper proposes FOMRC based on the Farrow structure FD filter to ensure that the resonant frequency of MRC can track the grid frequency in real time to obtain frequency adaptability.

The FOMRC system maintains a high open-loop gain at the fundamental and harmonic frequencies with variable grid frequency, allowing it to provide higher tracking progress and harmonic suppression performance, thus the system has a higher tracking gain and a lower THD.

What's more, by setting the sampling factor to 2, the sampling frequency of the repetitive controller part is reduced to half of the original, which reduces the calculation. The simulation results verified the effectiveness of the proposed FOMRC scheme in terms of fast transient response, high tracking accuracy, and immunity to frequency variations.

**Author Contributions:** Conceptualization, K.L.; methodology, H.L. and Q.Z.; software, K.L.; valida­tion, K.L.; formal analysis, K.L. and Q.Z.; writing—original draft preparation, K.L.; writing—review and editing, K.L., H.L. and Q.Z.; visualization, K.L.; supervision, H.L. and Q.Z.; project administra­tion, Q.Z.; funding acquisition, Q.Z. All authors have read and agreed to the published version of the manuscript.

**Funding:** This research was funded by the National Natural Science Foundation of China (No. 61973157, No. 62073297), and the Incubation program for young master supervisors of Zhongyuan University of Technology (No. D202213).

**Data Availability Statement:** The authors confirm that the data supporting the findings of this study are available within the article.

**Conflicts of Interest:** The authors declare no conflict of interest.

## Abbreviations

The following abbreviations are used in this manuscript:

| | |
|---|---|
| MRC | Multi-rate Repetitive Controller |
| CRC | Conventional Repetitive Controller |
| FOMRC | Fractional Order Multi-rate Repetitive Controller |
| FD | Fractional Delay |
| THD | Total Harmonic Distortion |
| PI | Proportional Integral |
| PR | Proportional Resonant |
| PMR | Proportional Multi-resonant |
| RC | Repetitive Controller |
| FIR | Finite Impulse Response |

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
