# Peer review of "A Fractional-Order Multi-Rate Repetitive Controller for Single-Phase Grid-Connected Inverters"

_electronics, doi:10.3390/electronics12041021_

Round 1

Reviewer 1 Report

The paper is well written and prents a good approach;

However, your objective is to reduce the calculation time so why you use a fractionnal order controller wich needs computing time, why do not use an other structure?

Reviewer 2 Report

Review: Electronics-2163727

The manuscript Electronics-2163727 reports the "A Fractional-order Multi-rate Repetitive Controller for Single-phase Grid-connected Inverters". This work shows an application about a repetitive controller for single-phase inverters. Work reports an average revision, but the problem characterization is well reported by authors. Therefore, the manuscript needs of minor reviews before publishing. Thus, I am suggesting some main points

1.    The abstract must be improved. Authors must provide more details of work results in abstract

2.    The review of the literature needs be improved by authors.

3.    The text of work shows moderate English. However, I advise to authors that check the grammatical part of English in paper. After reviewing, the article can be considered for a possible publication.

4.    The novelty of article needs to be detailed in introduction of work. Some novel points can be found in the manuscript body, and then this can be used to describe the novelty.

5.    In the conclusion, authors need to discuss the proposed theme with its innovation proposal.

Author Response

Please see the attachment. Revisions are highlighted in yellow in the second half of the updated manuscript (supplementary material for review).

Reviewer 3 Report

In this manuscript, the authors propose a fractional order multi-rate repetitive controller with a Farrow structure fractional delay filter based on Taylor series expansion to prevent the degeneration of harmonic suppression performance. Based on the simulation result, the Farrow structure fractional delay based on Taylor series expansion can effectively downgrade total harmonic distortion. Overall, this is a well-organized study on the topic of using simulation to optimize the steady-state accuracy of multi-rate repetitive controller when the grid frequency fluctuates. The reviewer has below concerns for authors to answer before it can be published in Electronics.

1. In Figure 7, could the authors clarify what the arrows represent on the three curves and also add more explanations about the three curves in this figure?

2. In the conclusion, the authors should add concluded data to make the statement more convincing. How is the proposed strategy able to effectively adapt to the change of power grid frequency, ensure higher tracking accuracy and lower THD. The authors need to provide summarize data results corresponding to these three conclusions, respectively. 

Author Response

(The authors gave the same response as above.)
